# Analysis of retinal markers and incident amyotrophic lateral sclerosis: An optical coherence tomography-based cohort study

Chunyang Pang[1,2☯], Yaojia Li[1,2☯], Wenhua Jiang[1,2], Haobo Xie[1,2], Wen Cao[3], Huan Yu[4], Zhiyang Lin[5], Yifan Cheng[6], Dongsheng Fan[3], Binbin Deng[1,2*]

1 Department of Rehabilitation Medicine, First Affiliated Hospital of Wenzhou Medical University, Wenzhou, PR. China, 2 Department of Neurology, First Affiliated Hospital of Wenzhou Medical University, Wenzhou, PR. China, 3 Department of Neurology, Peking University Third Hospital, Beijing, PR. China, 4 Department of Pediatrics, Second Affiliated Hospital and Yuying Children's Hospital of Wenzhou Medical University, Wenzhou, PR. China, 5 Department of Ophthalmology, The Fifth Affiliated Hospital of Wenzhou Medical University, Lishui, PR. China, 6 Department of Neurology, Zhejiang Provincial People's Hospital, Hangzhou, PR. China

☯ These authors contributed equally to this work.
* dbinbin@aliyun.com

## Abstract

### Background

Biomarkers are widely recognized as crucial breakthroughs in tackling amyotrophic lateral sclerosis (ALS). Among them, retina markers may hold promise due to the close retina-brain connection and non-invasive, portable detection methods. Thus, using optical coherence tomography (OCT), we investigated the link between baseline cell-level retinal features and future ALS risk.

### Methods and findings

Participants from the UK Biobank underwent OCT scans to assess retinal layers, macula, and optic disc parameters. Follow-up commenced two years after the baseline period (2006–2010), during which ALS cases were identified using International Classification of Diseases (ICD) codes from medical and assessment records. Cox proportional hazards models were applied to examine the relationship between retinal markers and incident ALS. Over a median follow-up of 14.11 years, 70 ALS cases occurred among 53,824 participants (incidence 10.58 per 100,000 person-years). Most participants were White (94.6%), 44.8% male, with a median age of 58 years. After adjusting for demographics and comorbidities affecting the retina, a standard deviation (SD) decrease of 15.19 μm in photoreceptor layer (PRL) thickness was associated with a 19% (95% confidence interval [7, 29]; $p = 0.002$) increased risk of ALS, while a SD increase of 26.11 μm in retinal pigment epithelium (RPE) thickness corresponded to a 20% (95% CI [7, 34]; $p = 0.002$) higher risk. Sensitivity analyses

**Data availability statement:** The data supporting the findings of this study are available through the UK Biobank project site and require successful registration and application due to ethical and legal restrictions protecting participant privacy and data security. Further details on access procedures can be found at https://www.ukbiobank.ac.uk. Our research data derived from the UK Biobank dataset are available from the UK Biobank upon reasonable request. For data access inquiries, please contact the UK Biobank directly via their website or email: access@ukbiobank.ac.uk.

**Funding:** This research was supported by the National Natural Science Foundation of China (Grant No. 81901273 to BD, http://www.nsfc.gov.cn) and the Natural Science Foundation of Zhejiang Province (Grant No. ZCLY24H0903 to BD, http://www.zjkjt.gov.cn). The funders had no role in study design, data collection and analysis, decision to publish, or preparation of the manuscript.

**Competing interests:** The authors have declared that no competing interests exist.

**Abbreviations:** AD, Alzheimer's disease; ALS, amyotrophic lateral sclerosis; BMI, body mass index; FTD, frontotemporal dementia; GCIPLT, Ganglion cell-inner plexiform layer thickness; HR, hazard ratio; ICD, International Classification of Diseases; INL, inner nuclear layer; IQR, interquartile range; MSA, multiple system atrophy; CT, optical coherence tomography; OR, odds ratio; PD, Parkinson's disease; PRL, photoreceptor layer; PSP, progressive supranuclear palsy; RNFL, retinal nerve fiber layer; ROS, reactive oxygen species; RPE, retinal pigment epithelium; SD, standard deviation; STROBE, Strengthening the Reporting of Observational Studies in Epidemiology; TDI, Townsend Deprivation Index; VCDR, vertical cup-to-disc ratio.

excluding follow-ups of less than 4 and 6 years yielded consistent results. Subgroup analyses showed these findings were more pronounced in smokers. The main limitation of this study is its single time point observational design.

## Conclusion

A thinner PRL and thicker RPE may precede the clinical diagnosis of ALS, offering potential clues for early diagnosis and insights into the disease's pathogenesis.

---

## Author summary

### Why was this study done?

- Amyotrophic lateral sclerosis (ALS) has limited diagnostic and treatment options, creating an urgent need for biomarkers to advance management.
- Retinal biomarkers may be emerging for ALS diagnosis due to retina-brain links and non-invasive detection.
- Current evidence on early retinal features in ALS is limited.

### What did the researchers do and find?

- Cell-level retinal features were assessed using optical coherence tomography (OCT) data from 53,824 participants.
- Their association with future ALS risk was investigated over a median follow-up of 14.11 years.
- A thinner photoreceptor layer (PRL) and a thicker retinal pigment epithelium (RPE) were found to potentially precede the clinical diagnosis of ALS.

### What do these findings mean?

- OCT-based assessment of PRL and RPE thickness may offer valuable clues for early ALS diagnosis and insights into its pathogenesis.
- The main limitation of this study is its single time point observational design.

## Introduction

Amyotrophic lateral sclerosis (ALS) is characterized by the relentless degeneration of neurons responsible for voluntary muscle movement, with an incidence of approximately 1–3 new cases per 100,000 people annually [1]. This disease remains fatal due to unclear causes and lack of effective treatments, with patients typically dying within three years, mainly from respiratory failure [2]. A hypothesis suggests that ALS, like other neurodegenerative diseases, may be preceded by a prolonged asymptomatic phase, supported by growing in vitro and animal evidence [3]. However, due to

the disease's rarity and heterogeneous presentations, diagnostic delay remains a challenge for ALS, let alone identifying preclinical stages [4]. Thus, there is an urgent need for objective, quantifiable biomarkers to enable early detection, enhance our understanding of ALS pathogenesis, and drive the development of new therapies [5].

There has been growing interest recently in using the eye to detect ALS, with the retina potentially offering an enticing possibility [6]. The eye, originating from the forebrain during embryonic development, shares structural similarities with the brain and may have common physiological processes relevant to ALS [7]. For example, optineurin, TANK-binding kinase 1, and ataxin-2 are linked to both familial ALS and primary open-angle glaucoma [8,9], and a unique association has been found between linear retinal pigment epitheliopathy and ALS/parkinsonism-Dementia complex [10]. On the other hand, ALS is increasingly recognized as a multisystem disorder [11], involving not only the motor system but also the ocular system, including abnormalities in oculomotor function, visual pathways, and the retina [12]. Some human pathological studies in patients with ALS have observed the accumulation of disease-specific inclusions in the inner plexiform layer [13], along with ganglion cell axonal loss [14], indicating pathological ocular involvement in ALS. Considering the potential link between the eye and ALS, the eye could be a candidate for exploration in large-scale studies to identify diagnostic and prognostic biomarkers for ALS [15]. Leveraging the eye's transparent optical media and advancements in high-resolution imaging, particularly Optical Coherence Tomography (OCT), retinal morphology can now be accessed quickly and non-invasively, reaching the cellular level [16,17]. Retinal correlates have been identified in a range of neurodegenerative conditions, including Alzheimer's disease (AD) and Parkinson's disease (PD), suggesting that neurodegenerative changes may be reflected in the retina [18]. If feasible, using retinal biomarkers in vivo as an additional diagnostic tool may aid in ALS risk stratification and provide a valuable opportunity for longitudinal monitoring over time.

Despite growing attention, studies on retinal architectural changes yielded contradictory and limited findings. Some studies suggest retinal layer thinning in patients with ALS [19,20], which aligns with the interpretation of neurodegenerative changes, while others report thickening of certain retinal layers, attributing it to inflammation [21]. For example, while some studies report reduced retinal nerve fiber layer (RNFL) thickness [22], which is widely recognized as a marker of retinal ganglion cell loss [23] and considered an indicator of neurodegeneration, a meta-analysis of 11 studies found no significant difference in RNFL thickness between patients with ALS and healthy controls [24]. It is important to note that the majority of these studies are small-sample, cross-sectional case-control studies, many of which do not account for confounding factors such as age, refractive status, and disease duration [25]. Additionally, there is a lack of prospective cohort studies examining OCT data in relation to ALS risk, which could potentially reveal early features of the disease, thus guiding earlier intervention and research with greater clinical relevance.

In this study, we employed the large UK Biobank cohort to examine the association between OCT retinal imaging data and the onset of ALS. Our objective was to identify significant retinal changes during the pre-diagnose phase of ALS, with the ultimate goal of providing additional evidence for the use of retinal markers in ALS diagnosis and progression.

## Methods

### Data source and study population

The research utilized prospective data from the UK Biobank Resource (Application Number 108832), with written informed consent from all participants and ethical approval from the Northwest-Multicenter Research Ethical Committee. Between 2006 and 2010, UK Biobank recruited over 500,000 participants from the general population, obtaining their clinical, biochemical, and eye measures, among other information [26]. As shown in the diagram (Fig 1), a subset of 67,522 UK Biobank participants underwent a detailed eye assessment (initial assessment), including visual acuity, auto-refraction, intraocular pressure, and OCT scan. After excluding those with ALS at baseline ($n = 6$), those with follow-up of less than 2 years ($n = 323$), and the worst 20% of images based on specific image quality control metadata ($n = 13,444$), a total of 53,824 participants were included in the prospective study. This study is reported as per the Strengthening the Reporting of Observational Studies in Epidemiology (STROBE) guideline (S1 STROBE Checklist).

PLOS Medicine

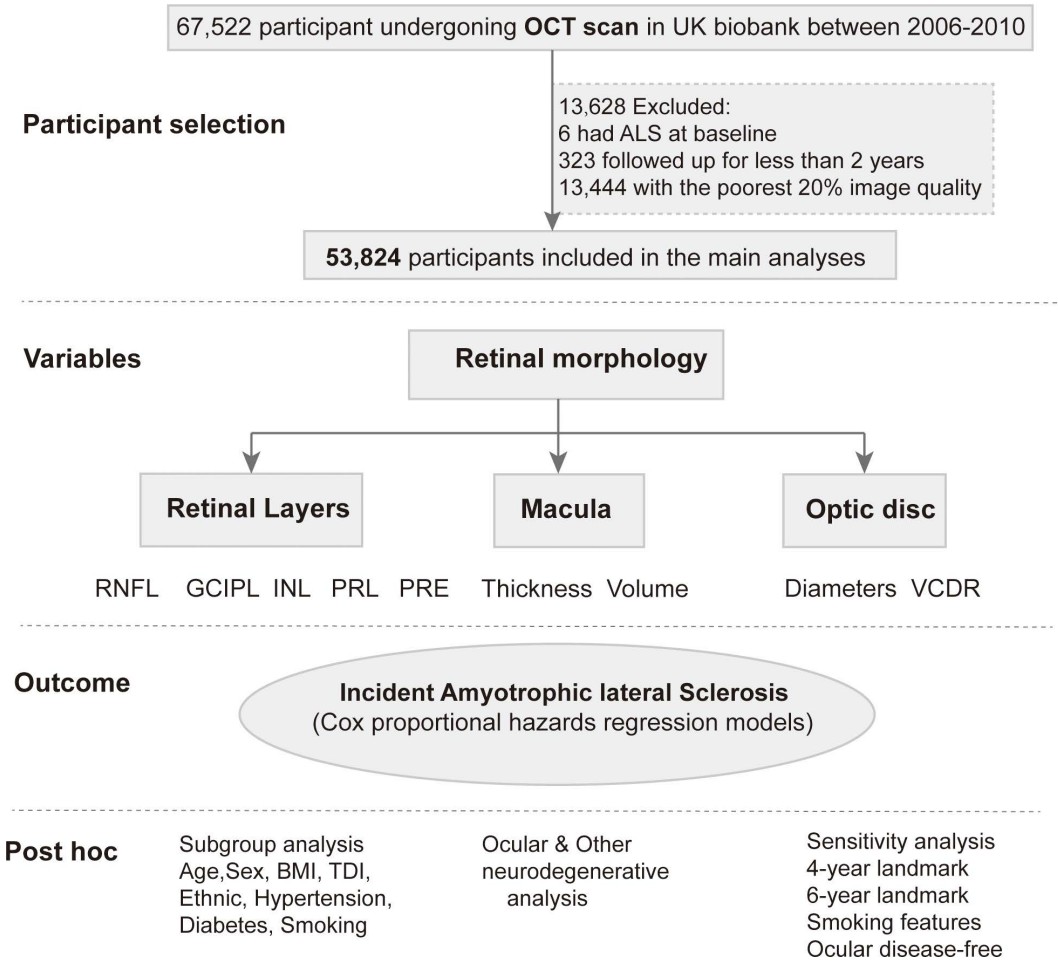

**Fig 1. Participant selection and study design.** Abbreviations: OCT, Optical coherence tomography; TDI, Townsend Deprivation Index; BMI, Body mass index; RNFL, Retinal nerve fiber layer; GCIPLT, Ganglion cell-inner plexiform layer thickness; INL, Inner nuclear layer; PRL, Photoreceptor layer; RPE, Retinal pigment epithelium; VCDR, Vertical cup-to-disc ratio.

## ALS diagnosis and ocular diseases

ALS were derived by mapping Read codes from Primary Care data (Category 3000), International Classification of Diseases (ICD) codes from Hospital inpatient data (Category 2000), ICD-10 codes from Death Register records (Fields 40001, 40002), and self-reported medical condition codes (Field 20002) from baseline or subsequent UK Biobank assessment center visits to 3-character ICD-10 codes, along with the date when the code was first recorded across any of the listed sources [27,28]. In the UK Biobank, only the parent codes for ALS are used in routine clinical practice, including ICD-9 code 335.2 and ICD-10 code G12.2. Additionally, 36 ocular diseases were included in the analysis, with detailed code information summarized in the Table A in S1 Text.

## OCT scan

OCT scans were performed during the initial assessment visit, which took place between 2006 and 2010 (baseline period), provided that participants had not undergone any eye surgeries or experienced eye infections in the preceding four weeks. Participants sat in a closed, dark room with no glare, maintaining the correct humidity and temperature to ensure the effective

operation of the eye measurement equipment. The TOPCON 3D OCT 1000 Mk2 (Topcon Corporation, Tokyo, Japan) is used at the UK Biobank Assessment Centre, and retinal measurements are estimated using the Topcon Advanced Boundary Segmentation Tool (TABS) version 1.6.2.6 for automated sublayer segmentation [29]. Five retinal layer indicators corresponding to the distribution of cell types in the retina were included: RNFL, ganglion cell layer, inner plexiform layer (IPL), inner nuclear layer (INL), photoreceptor layer (PRL), and retinal pigment epithelium (RPE), along with macular thickness, macular volume, disc diameters, and vertical cup-to-disc ratio (VCDR). The average value of the data from both eyes was included in the calculation, and if data from only one eye was available, the data from that eye was used.

## Statistical analysis

Baseline characteristics are presented as median and interquartile range (IQR) for continuous variables, while categorical variables are summarized as counts and percentages. OCT parameters were standardized using Z-scores to ensure comparability across variables, as done in previous studies [30,31].

All statistical analyses were performed using R software, version 4.2.1. The figures were created in Adobe Illustrator. A two-sided p-value less than 0.05 was considered significant; after Bonferroni correction, the significance threshold was adjusted to $p < 0.0056$.

Cox proportional hazards models were used to investigate the association between OCT parameters and incident ALS. Participants were censored at the date of ALS diagnosis, death, or the end of follow-up, whichever occurred first. Non-linear relationships (p for non-linearity) were assessed using restricted cubic splines with three knots placed at the 10th, 50th, and 90th percentiles. Covariates, including age, sex, ethnicity, Townsend Deprivation Index (TDI), hypertension, diabetes, intraocular pressure, and spherical equivalent, were included (Methods in S1 Text). Proportional hazard assumptions were evaluated using Schoenfeld residuals for all models (Table B in S1 Text). The multicollinearity analyses were assessed and showed Variance inflation factor <2.5 for independent covariates (Table C in S1 Text).

In the post hoc analysis (Fig 1), subgroup analyses were conducted, stratified by age, sex, ethnicity, TDI, hypertension, diabetes, body mass index (BMI), and smoking status, to investigate population-specific effects. Second, the association between OCT parameters and other neurodegenerative diseases was explored to assess the specificity of these markers. Logistic regression were performed to examine the relationship between baseline ocular diseases and the risk of developing ALS, excluding individuals diagnosed during the follow-up period. Additionally, sensitivity analyses were performed by excluding individuals with follow-ups shorter than 4–6 years, removing individuals with ocular diseases, and adjusting for smoking-related factors to minimize potential confounding effects and reduce the risk of reverse causality.

## Results

### Characteristics of study participants

A total of 53,824 participants were included in this study. Over a median follow-up of 14.11 years, 70 ALS cases were identified, with an incidence rate of 10.58 per 100,000 person-years. As shown in Table 1, the majority of included participants were of White ethnicity (94.6%), and 44.8% were male, with a medium age of 58 years. There was a notable prevalence of hypertension (26.0%) and some prevalence of diabetes (5.1%). Among the subjects included and excluded from the study, significant differences were observed in sex ($p < 0.001$), BMI ($p = 0.043$), hypertension ($p < 0.001$), and diabetes ($p < 0.001$). Besides, in the included participants, the median spherical equivalent was 0.18 diopters, with an IQR of [−0.85, 1.09] diopters, and the median intraocular pressure was 15.62 mmHg, with an IQR of [13.52, 17.89] mmHg.

### Distribution of OCT parameters and their nonlinear relationship with incident ALS risk

The distribution of OCT parameters is shown as a histogram (Fig A in S1 Text), revealing that most variables were skewed. Normality tests showed that none of these variables had a $p > 0.05$ in both the Kolmogorov–Smirnov and

**Table 1. General characteristics of the study population.**

| | Overall | Included | Excluded |
|---|---|---|---|
| Sample size | 67,522 | 53,824 | 13,698 |
| **Baseline information** | | | |
| Age, years | 58.00 [50.00, 63.00] | 58.00 [50.00, 63.00] | 58.00 [50.00, 63.00] |
| Sex, male | 30,800 (45.6) | 24,123 (44.8) | 6,677 (48.7) |
| Ethnicity | | | |
| Black | 1,124 (1.7) | 896 (1.7) | 228 (1.7) |
| Other/mix | 1,273 (1.9) | 1,014 (1.9) | 259 (1.9) |
| South Asian | 1,269 (1.9) | 1,001 (1.9) | 268 (2.0) |
| White | 63,496 (94.5) | 50,636 (94.6) | 12,860 (94.5) |
| TDI | −2.16 [−3.65, 0.53] | −2.15 [−3.65, 0.52] | −2.17 [−3.65, 0.53] |
| Smoking history | | | |
| Never | 36,787 (54.8) | 29,357 (54.8) | 7,430 (54.6) |
| Previous | 23,354 (34.8) | 18,571 (34.7) | 4,783 (35.1) |
| Current | 7,012 (10.4) | 5,613 (10.5) | 1,399 (10.3) |
| BMI, Kg/m2 | 26.76 [24.14, 29.94] | 26.74 [24.14, 29.91] | 26.85 [24.17, 30.00] |
| Hypertension, yes | 17,894 (26.5) | 13,979 (26.0) | 3,915 (28.6) |
| Diabetes, yes | 3,668 (5.4) | 2,762 (5.1) | 906 (6.6) |
| Spherical equivalent, diopters | 0.10 [−1.21, 1.05] | 0.18 [−0.85, 1.09] | −0.49 [−3.27, 0.81] |
| IOP, mmHg | 15.67 [13.56, 17.94] | 15.62 [13.52, 17.89] | 15.86 [13.71, 18.19] |
| **Retinal morphology** | | | |
| RNFL, micrometers | 27.92 [25.09, 30.94] | 28.24 [25.58, 31.11] | 26.40 [22.78, 30.04] |
| GCIPLT, micrometers | 73.78 [69.67, 77.74] | 74.16 [70.25, 77.92] | 71.98 [67.08, 76.61] |
| INL, micrometers | 32.44 [30.85, 34.12] | 32.33 [30.77, 33.92] | 32.96 [31.20, 35.09] |
| PRL, micrometers | 141.96 [136.59, 147.35 | ] 141.54 [136.40, 146.82] | 143.72 [137.62, 149.48] |
| RPE, micrometers | 24.86 [23.36, 26.98] | 24.91 [23.40, 27.00] | 24.67 [23.21, 26.89] |
| Macular thickness, micrometers | 276.52 [266.94, 285.96] | 276.72 [267.48, 285.90] | 275.48 [264.54, 286.25] |
| Macular volume, mm3 | 7.87 [7.63, 8.12] | 7.87 [7.63, 8.11] | 7.88 [7.63, 8.13] |
| Disc diameter, micrometers | 121.07 [111.36, 131.06] | 121.26 [112.01, 131.10] | 120.04 [110.00, 131.00] |
| VCDR, ratio | 0.30 [0.20, 0.40] | 0.30 [0.20, 0.40] | 0.30 [0.20, 0.45] |

**Abbreviations:** TDI, Townsend Deprivation Index; BMI, Body mass index; IOP, Intraocular pressure; RNFL, Retinal nerve fiber layer; GCIPLT, Ganglion cell-inner plexiform layer thickness; INL, Inner nuclear layer; PRL, Photoreceptor layer; RPE, Retinal pigment epithelium; SD, Standard deviation; VCDR, Vertical cup-to-disc ratio.

Anderson–Darling tests (Table D in S1 Text), indicating a deviation from a normal distribution. As shown in Fig 2, a significant decreasing linear association was observed between PRL and the risk of incident ALS ($p$ for nonlinearity = 0.876), whereas a linear increase was observed for RPE ($p$ for nonlinearity = 0.170). Moreover, a U-shaped association between INL and incident ALS risk was observed, but it was not statistically significant ($p$ for nonlinear = 0.270).

## Cox analysis of retinal markers and incident ALS risk

The results of the Cox analysis are summarized in Table 2. In the crude model, elevated INL (per SD increase hazard ratio (HR) [95% CI] = 1.17 [1.06, 1.29]; $p$ = 0.003) and RPE (per SD increase HR [95% CI] = 1.20 [1.07, 1.34]; $p$ = 0.002) were associated with an increased risk of ALS, while elevated PRL (per SD increase HR [95% CI] = 0.82 [0.71, 0.94]; $p$ = 0.004) was associated with a reduced risk. In the multivariable Cox analysis, adjusted for age, sex, ethnicity, TDI, hypertension,

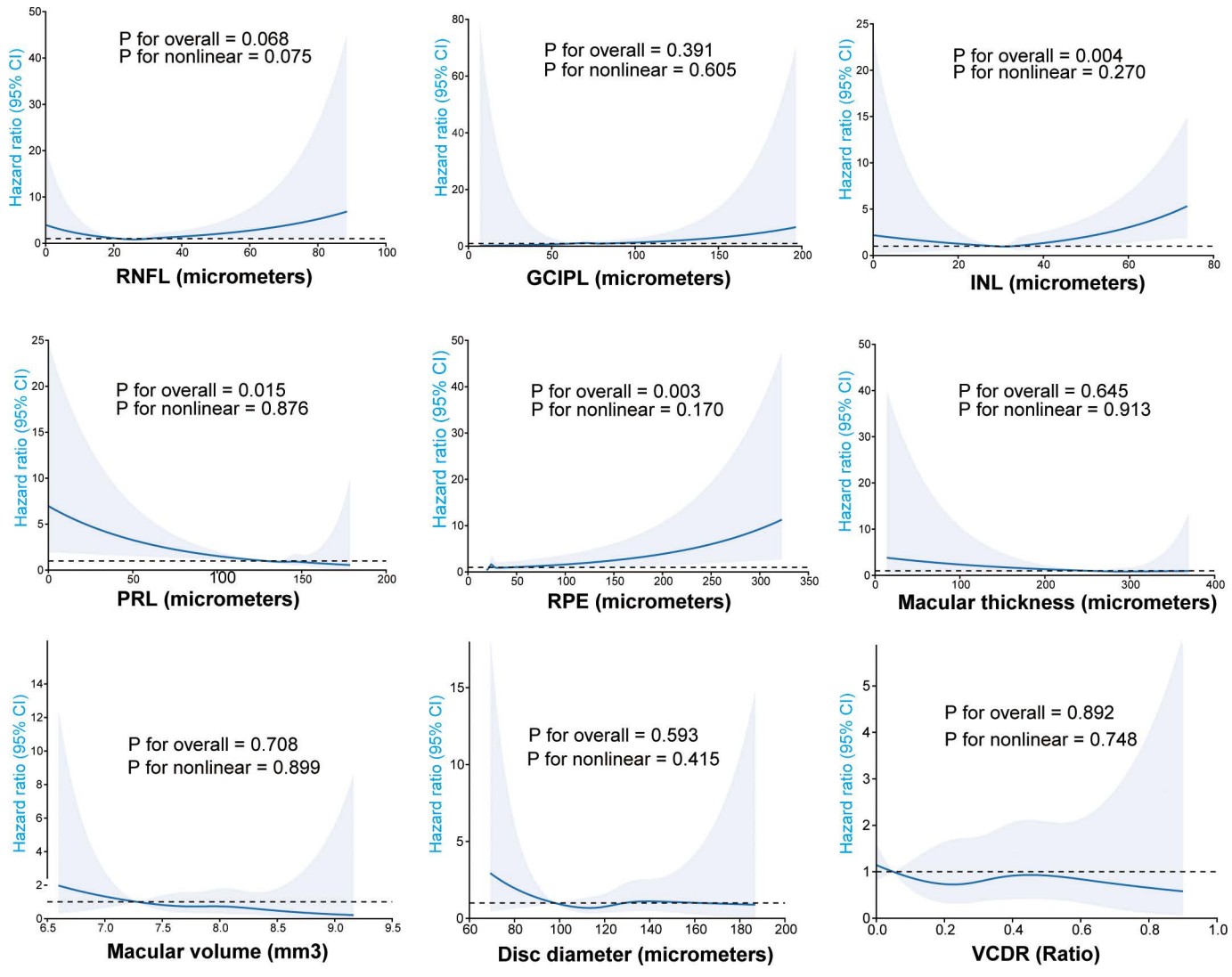

**Fig 2. Non-linear relationship retinal markers with incident ALS over a median follow-up of 14.11 years.** Adjusted for age, sex, ethnicity, TDI, hypertension, diabetes, intraocular pressure, and spherical equivalent. Shaded area represents the 95% confidence interval. **Abbreviations:** TDI, Townsend Deprivation Index; BMI, Body mass index; RNFL, Retinal nerve fiber layer; GCIPLT, Ganglion cell-inner plexiform layer thickness; INL, Inner nuclear layer; PRL, Photoreceptor layer; RPE, Retinal pigment epithelium; SD, Standard deviation; VCDR, Vertical cup-to-disc ratio.

diabetes, intraocular pressure, and spherical equivalent, the conclusions remain consistent: INL (per SD increase HR [95% CI] = 1.16 [1.04, 1.29]; $p = 0.009$), RPE (per SD increase HR [95% CI] = 1.20 [1.07, 1.34]; $p = 0.002$), and PRL (per SD increase HR [95% CI] = 0.81 [0.71, 0.93]; $p = 0.002$). Besides, after Bonferroni correction, the results for PRL and RPE remained significant ($p < 0.0056$). However, there were no associations between RNFL, GCIPL, macular thickness, macular volume, disc diameter, VCDR, and ALS risk.

In the sensitivity analysis, after excluding individuals with a follow-up of less than 4 years, a thinner PRL (per SD increase HR [95% CI] = 0.84 [0.72, 0.99]; $p = 0.040$) and a thicker RPE (per SD increase HR [95% CI] = 1.16 [1.01, 1.33]; $p = 0.041$) was still associated with higher ALS risk, while INL was not (per SD increase HR [95% CI] = 1.05 [0.79, 1.38]; $p = 0.741$). After excluding individuals with a follow-up of less than 6 years, PRL (per SD increase HR [95% CI] =

**Table 2. Associations between retinal markers and incident ALS.**

| Incident ALS | Models | Per SD increase HR (95%CI) | p |
|---|---|---|---|
| RNFL (mm) | Unadjusted | 1.10 (0.82,1.48) | 0.536 |
| | Adjusted | 1.16 (0.88,1.53) | 0.308 |
| | Sensitivity analysis 1 | 1.24 (0.98,1.57) | 0.077 |
| | Sensitivity analysis 2 | 1.13 (0.80,1.58) | 0.496 |
| GCIPL (mm) | Unadjusted | 1.13 (0.94,1.34) | 0.187 |
| | Adjusted | 1.10 (0.90,1.35) | 0.338 |
| | Sensitivity analysis 1 | 1.10 (0.89,1.37) | 0.377 |
| | Sensitivity analysis 2 | 1.10 (0.87,1.40) | 0.428 |
| INL (mm) | Unadjusted | **1.17 (1.06,1.29)** | **0.003 *** |
| | Adjusted | **1.16 (1.04,1.29)** | **0.009** |
| | Sensitivity analysis 1 | 1.05 (0.79,1.38) | 0.741 |
| | Sensitivity analysis 2 | 1.02 (0.74,1.40) | 0.916 |
| PRL (mm) | Unadjusted | **0.82 (0.71,0.94)** | **0.004 *** |
| | Adjusted | **0.81 (0.71,0.93)** | **0.002 *** |
| | Sensitivity analysis 1 | **0.84 (0.72,0.99)** | **0.040** |
| | Sensitivity analysis 2 | **0.82 (0.70,0.97)** | **0.018** |
| RPE (mm) | Unadjusted | **1.20 (1.07,1.34)** | **0.002 *** |
| | Adjusted | **1.20 (1.07,1.34)** | **0.002 *** |
| | Sensitivity analysis 1 | **1.16 (1.01,1.33)** | **0.041** |
| | Sensitivity analysis 2 | **1.19 (1.03,1.37)** | **0.015** |
| Macular thickness (mm) | Unadjusted | 0.90 (0.73,1.11) | 0.332 |
| | Adjusted | 0.89 (0.73,1.08) | 0.239 |
| | Sensitivity analysis 1 | 0.93 (0.73,1.18) | 0.533 |
| | Sensitivity analysis 2 | 0.88 (0.70,1.10) | 0.249 |
| Macular volume (mm³) | Unadjusted | 0.83 (0.59,1.17) | 0.285 |
| | Adjusted | 0.82 (0.58,1.17) | 0.279 |
| | Sensitivity analysis 1 | 0.82 (0.58,1.17) | 0.284 |
| | Sensitivity analysis 2 | 0.80 (0.53,1.19) | 0.270 |
| Disc diameter (mm) | Unadjusted | 0.97 (0.75,1.26) | 0.842 |
| | Adjusted | 0.92 (0.70,1.20) | 0.526 |
| | Sensitivity analysis 1 | 0.89 (0.67,1.18) | 0.417 |
| | Sensitivity analysis 2 | 1.08 (0.80,1.46) | 0.627 |
| VCDR (ratio) | Unadjusted | 0.93 (0.72,1.21) | 0.594 |
| | Adjusted | 0.94 (0.72,1.23) | 0.660 |
| | Sensitivity analysis 1 | 0.93 (0.70,1.22) | 0.589 |
| | Sensitivity analysis 2 | 0.97 (0.72,1.31) | 0.862 |

Adjusted for age, sex, ethnicity, TDI, hypertension, diabetes, intraocular pressure, and spherical equivalent; Sensitivity analysis 1: Excluding participants with a follow up less than 4 years; Sensitivity analysis 2: Excluding participants with a follow up less than 6 years.

*Significant after Bonferroni correction; Bold values indicate statistical significance at $p < 0.05$.

**Abbreviations:** ALS, Amyotrophic lateral sclerosis; TDI, Townsend Deprivation Index; BMI, Body mass index; IOP, Intraocular pressure; RNFL, Retinal nerve fiber layer; GCIPLT, Ganglion cell-inner plexiform layer thickness; INL, Inner nuclear layer; PRL, Photoreceptor layer; RPE, Retinal pigment epithelium; SD, Standard deviation; VCDR, Vertical cup-to-disc ratio; mm, micrometers.

0.82 [0.70, 0.97]; *p* = 0.018) and RPE (per SD increase HR [95% CI] = 1.19 [1.03, 1.37]; *p* = 0.015) remained significantly associated with ALS. Further analysis adjusting for smoking-related factors revealed consistent trends: after adjusting for smoking status (Table E in S1 Text), PRL was associated with a decreased risk of ALS (per SD increase HR [95% CI] = 0.81 [0.71, 0.93]; *p* = 0.008), while RPE was linked to an increased risk (per SD increase HR [95% CI] = 1.20 [1.07, 1.35]; *p* = 0.002). Similarly, adjusting for maternal smoking around birth (Table F in S1 Text) yielded comparable results for PRL (per SD increase HR [95% CI] = 0.81 [0.71, 0.93]; *p* = 0.002) and RPE (per SD increase HR [95% CI] = 1.20 [1.07, 1.34]; *p* = 0.002). Besides, a sensitivity analysis excluding individuals with ocular disease at baseline or during follow-up (*n* = 15,617) confirmed consistent results for PRL (per SD increase HR [95% CI] = 0.76 [0.66, 0.88]; *p* < 0.001) and RPE (per SD increase HR [95% CI] = 1.27 [1.13, 1.43]; *p* < 0.001) (Table G in S1 Text).

### Subgroup analysis and other disease analysis

In the subgroup analyses (Fig 3 and Table H in S1 Text), the result for PRL was more striking in white ethnicity (*p* for interaction = 0.027) and among participants who smoked (*p* for interaction = 0.018). However, since over 90% of the UK Biobank population is White, the sample size distribution in the ethnicity group was highly uneven (2,809 versus 48,890); therefore, the subgroup analysis stratified by ethnicity is unreliable. A for RPE, the results were also more striking in the smoking group (*p* for interaction = 0.024).

Relationship between OCT markers and five other neurodegenerative diseases were summarized in Table 3, The results show that changes in PRL and RPE may be specific to ALS: atrophy of the GCIPL, RNFL, macula, and optic disc may be more common in AD; a thicker GCIPL may be related to PD; a reduction in macular volume was observed in multiple system atrophy (MSA); atrophy of the PRL was noted in progressive supranuclear palsy (PSP). Besides, No significant

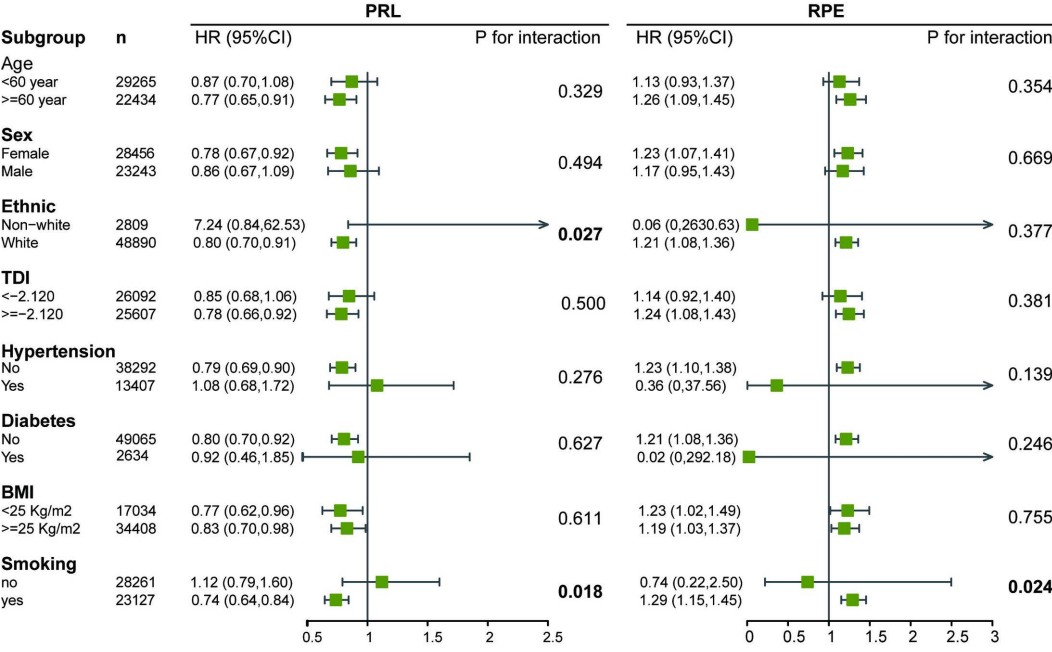

**Fig 3. Subgroup analysis for PRL and RPE.** Adjusted for age, sex, ethnicity, TDI, hypertension, diabetes, intraocular pressure, and spherical equivalent; TDI was dichotomized at the median value (2.120), with higher values indicating greater socioeconomic deprivation. Abbreviations: TDI, Townsend Deprivation Index; BMI, Body mass index; PRL, Photoreceptor layer; RPE, Retinal pigment epithelium.

**Table 3. Associations between OCT parameters and neurodegenerative diseases.**

| Index | ALS | | AD | | FTD | |
|---|---|---|---|---|---|---|
| | (case/n = 70/53824) | | (case/n = 332/53511) | | (case = 20/53514) | |
| | HR (95%CI) | p | HR (95%CI) | p | HR (95%CI) | p |
| Disc diameter (mm) | 0.92 (0.70,1.20) | 0.526 | 1.03 (0.90,1.18) | 0.638 | 1.15 (0.68,1.93) | 0.599 |
| GCIPL (mm) | 1.10 (0.90,1.35) | 0.338 | **0.78 (0.69,0.87)** | **0.000** * | 0.95 (0.56,1.64) | 0.863 |
| PRL (mm) | **0.81 (0.71,0.93)** | **0.002** * | 0.95 (0.86,1.05) | 0.283 | 1.23 (0.92,1.66) | 0.166 |
| INL (mm) | **1.16 (1.04,1.29)** | **0.009** | 0.95 (0.84,1.06) | 0.352 | 0.94 (0.59,1.52) | 0.804 |
| RNFL (mm) | 1.16 (0.88,1.53) | 0.308 | **0.85 (0.73,1.00)** | **0.045** | 1.12 (0.63,1.98) | 0.694 |
| RPE (mm) | **1.20 (1.07,1.34)** | **0.002** * | 1.02 (0.91,1.14) | 0.732 | 0.05 (0.00,6.44) | 0.226 |
| Macular thickness (mm) | 0.89 (0.73,1.08) | 0.239 | **0.89 (0.81,0.97)** | **0.007** | 1.18 (0.76,1.83) | 0.470 |
| Macular volume (mm³) | 0.82 (0.58,1.17) | 0.279 | **0.77 (0.64,0.91)** | **0.003** * | 0.86 (0.38,1.96) | 0.719 |
| VCDR (ratio) | 0.94 (0.72,1.23) | 0.660 | **1.23 (1.08,1.40)** | **0.002** * | 0.82 (0.49,1.36) | 0.436 |
| | **PD** | | **MSA** | | **PSP** | |
| **Index** | (case/n = 328/53429) | | (case/n = 13/53515) | | (case/n = 22/53515) | |
| | HR (95%CI) | p | HR (95%CI) | p | HR (95%CI) | p |
| Disc diameter (mm) | 0.97 (0.84,1.11) | 0.622 | 0.51 (0.25,1.04) | 0.063 | 0.93 (0.54,1.60) | 0.785 |
| GCIPL (mm) | **0.87 (0.77,0.99)** | **0.037** | 0.66 (0.42,1.03) | 0.067 | 1.09 (0.71,1.68) | 0.702 |
| PRL (mm) | 0.94 (0.85,1.03) | 0.179 | 1.08 (0.57,2.03) | 0.819 | **0.80 (0.64,0.99)** | **0.042** |
| INL (mm) | 0.94 (0.84,1.05) | 0.273 | 0.89 (0.53,1.49) | 0.646 | 1.20 (0.99,1.45) | 0.063 |
| RNFL (mm) | 0.98 (0.84,1.15) | 0.808 | 0.62 (0.31,1.22) | 0.166 | 1.18 (0.72,1.93) | 0.504 |
| RPE (mm) | 0.99 (0.87,1.13) | 0.897 | 0.61 (0.03,14.73) | 0.763 | 1.21 (0.99,1.47) | 0.060 |
| Macular thickness (mm) | 0.91 (0.83,1.01) | 0.068 | 0.85 (0.57,1.27) | 0.419 | 0.86 (0.63,1.19) | 0.365 |
| Macular volume (mm³) | 0.88 (0.74,1.05) | 0.151 | **0.36 (0.13,0.99)** | **0.047** | 0.88 (0.39,1.98) | 0.754 |
| VCDR (ratio) | 1.04 (0.91,1.19) | 0.583 | 0.61 (0.31,1.18) | 0.139 | 0.60 (0.35,1.03) | 0.065 |

Adjusted for age, sex, ethnicity, TDI, hypertension, diabetes, intraocular pressure, and spherical equivalent; *significant after Bonferroni correction; Bold values indicate statistical significance at $p < 0.05$.

Abbreviations: ALS, Amyotrophic lateral sclerosis; AD, Alzheimer's disease; PD, Parkinson's disease; FTD, Frontotemporal dementia; MSA, Multiple system atrophy; PSP, Progressive supranuclear palsy; TDI, Townsend Deprivation Index; BMI, Body mass index; IOP, Intraocular pressure; RNFL, Retinal nerve fiber layer; GCIPLT, Ganglion cell-inner plexiform layer thickness; INL, Inner nuclear layer; PRL, Photoreceptor layer; RPE, Retinal pigment epithelium; SD, Standard deviation; VCDR, Vertical cup-to-disc ratio; mm, micrometers.

findings were observed in frontotemporal dementia (FTD), despite it being considered part of the same disease spectrum as ALS.

The results of the ocular disease analysis are presented as a heatmap in Fig B in S1 Text. Thinned PRL and thickened RPE are commonly observed in H35-other retinal disorders (PRL: Odds ratio (OR) [95% CI] = 0.93 [0.87, 0.99], $p = 0.024$; RPE: OR [95% CI] = 1.08 [1.03, 1.15], $p = 0.005$) and H52-disorders of refraction and accommodation (PRL: OR [95% CI] = 0.907 [0.83, 0.99], $p = 0.035$; RPE: OR [95% CI] = 1.11 [1.03, 1.19], $p = 0.010$). Considering baseline disease conditions and incident ALS risk, baseline H31-other disorders of the choroid may be associated with an increased future risk of ALS, with significant HRs at the 2-year (HR [95% CI] = 11.90 [4.45, 31.80]; $p < 0.001$), 4-year (HR [95% CI] = 13.57 [5.08, 36.30]; $p < 0.001$), and 6-year (HR [95% CI] = 16.41 [5.27, 51.14]; $p < 0.001$) landmarks. Detailed logistic regression results between PRL and RPE with ocular disease disorders can be found in Table J in S1 Text Detailed Cox regression results between ocular disease disorders and incident ALS risk can be found in Table K in S1 Text.

## Discussion

In this study based on the UK Biobank cohort with a median follow-up of over 14 years, three key findings emerged (Fig 4): First, per SD reduction in PRL thickness was strongly associated with a 19% increased risk of ALS, while per SD increase in RPE thickness corresponded to a 20% higher risk, with these associations consistently observed 2, 4, and 6 years prior to ALS diagnosis. Second, the relationship between PRL and RPE thickness and the incidence of ALS was more pronounced in smokers. Third, the occurrence of non-inflammatory choroidal diseases may also be associated with the development of ALS.

A thicker RPE and a thinner PRL may be attributed to oxidative stress and mitochondrial dysfunction, which have been proven to play a role in the pathogenesis [32]. The RPE is a layer of cells in the retina that absorbs light, protects the retina from damage, and aids in nutrient transport and waste removal [33], which contains a high density of mitochondria essential for energy meets [34]. While oxygen is utilized to produce adenosine triphosphate in mitochondria, a large amount of reactive oxygen species (ROS) can also be generated [34]. Excessive ROS can lead to oxidative stress

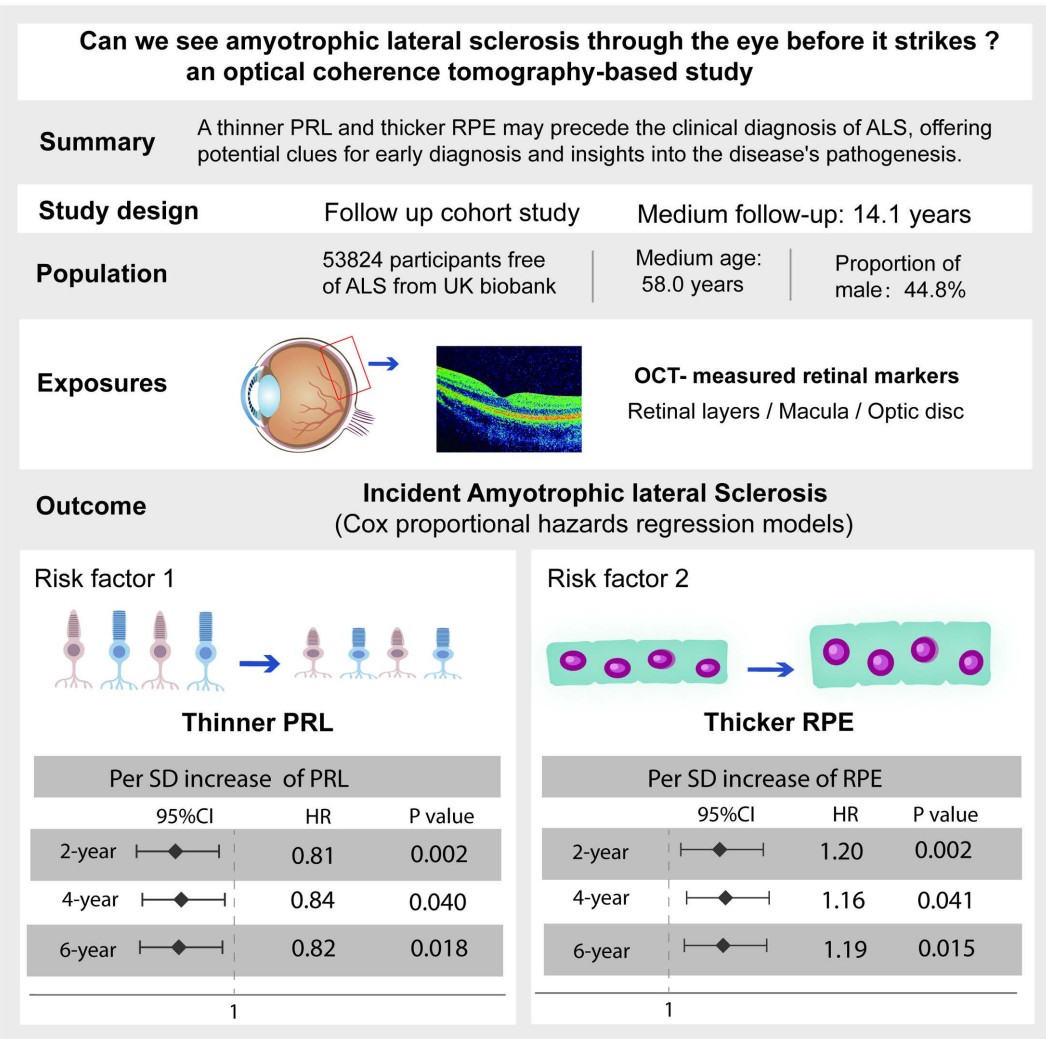

**Fig 4. Graphic abstract. Abbreviations:** OCT, Optical coherence tomography; PRL, Photoreceptor layer; RPE, Retinal pigment epithelium.

imbalance and mitochondrial dysfunction [32], often triggered by exogenous oxidative stress sources such as ultraviolet radiation, hyperglycemia, or cigarette smoke [35], or accumulated mitochondrial damage associated with aging [36]. Thus, pathological changes in the RPE are often observed in age-related macular degeneration [37] and diabetic retinopathy [38]. Interestingly, although RPE atrophy is commonly observed, our study found an increase in its thickness, which is rare and has only been reported in a few cases of prolonged disease duration in Parkinson's disease [39] and as a side effect of medications like pentosan polysulfate sodium [40]. Previous hypotheses have attributed RPE hypertrophy to cellular dysmorphia and a thickened basal lamina [41], potentially driven by oxidative stress, which may lead to progressive hypertrophy and eventual cell death [42]. A study using cryo-electron microscopy identified mitochondrial creatine kinase in a raw bovine RPE lysate [43], an enzyme that may help differentiate ALS from other neurodegenerative diseases, with both in vivo and in vitro data supporting its potential as a predisposing factor for disease progression and phenotype [44,45]. Therefore, it is reasonable to speculate that the process of RPE thickening may be associated with mitochondrial oxidative stress [42]. However, this appears to contradict the RPE atrophy observed in other conditions driven by similar oxidative stress mechanisms [37,38]. This discrepancy may reflect a unique oxidative stress dysregulation in ALS or different disease stages, such as early inflammation and edema followed by late-stage atrophy. On the other hand, the PRL is composed of rod cells and cone cells, responsible for converting light signals into neural signals [46], relies on the specialized and dynamic barrier functions of the RPE for structural and functional stability [47]. Photoreceptor cells are the most abundant cells in the retina [48] and among the most metabolically active cells in the body [49], making them highly susceptible to energy metabolism fluctuations. Our study suggests that PRL may undergo thinning in the preclinical stage of ALS, which is consistent with a previous study showing thinning of the photoreceptor cell nuclei [50] and also supports the finding of subclinical decline in visual acuity in patients with ALS [51]. However, due to limited evidence, the explanation for PRL thinning remains unclear. Based on the strong negative correlation between PRL and RPE thickness observed in Analysis A in S1 Text, we speculate that mitochondrial dysfunction and oxidative stress in the RPE may contribute to the degeneration of energy-dependent photoreceptors. Further studies are needed to elucidate the histopathological and molecular alterations of RPE and PRL in ALS.

Although there may be potential consistency in the pathogenesis, are RPE and PRL part of ALS disease?

On the one hand, our study suggests that retinal changes may reflect the combined effects of multiple ALS-related pathogenic factors. The pathogenesis of ALS is unclear but likely involves genetic and multiple environmental factors [52]. Cigarette smoking has long been proposed as a risk factor for ALS [53], with a strong causal relationship suggested [54], potentially contributing to ALS through lipid peroxidation [55], which transforms oxidative stress into a pathological factor that disrupts the protective roles of mitochondrial dynamics and the Nrf2 signaling pathway, ultimately compromising retinal health [56]. Therefore, the retina may serve as a reflection of smoking-related effects. Besides, further sensitivity analyses revealed that after adjusting for smoking-related factors and excluding individuals with ocular diseases, the associations remained statistically significant, suggesting the presence of other ALS-specific pathogenic factors similar to smoking. On the other hand, our study suggests that retinal changes may have specificity. We found that PRL remained significantly positively associated with the volume of gray matter in the precentral gyrus (S4 Analysis D in S1 Text), while gray matter atrophy in this region has previously been implicated in ALS [30,57]. Furthermore, cross-disease studies suggest that the early retinal changes we observed in ALS were not present in diseases such as AD and PD. However, due to the difficulty of obtaining human tissue in the early stages of the disease, evidence for most early characteristics of ALS remains limited. Future studies are needed to further determine whether the retinal changes we observed are specific biomarkers of ALS and to explore their correlation with factors such as TAR DNA-binding protein 43 [58].

Retinal vascular alternations may also be involved in ALS. Changes in the small blood vessels of the skin, muscles, and brain have been reported in patients with ALS [59,60], indicating the rationale for considering retinal microvasculature as a potential mirror for this disease. In this study, we found that non-inflammatory choroidal diseases may be linked to future incident ALS onset. The choroid, located outside the RPE, supplies oxygen, nutrients, and removes waste to

maintain retinal health. Previous studies have found choroidal thinning in PD [61], but evidence linking choroidal diseases to neurodegenerative diseases is limited. Non-inflammatory choroidal diseases are disorders that usually involve changes in blood flow and vascular abnormalities [62], which may be consistent with previous studies reporting retinal vascular alterations in ALS. However, to date, research on choroidal and retinal vascular changes in ALS remains a relatively novel field with limited evidence, highlighting the need for further studies to substantiate these findings.

Previous OCT studies exploring biomarkers for ALS have primarily focused on the post-diagnosis stage; to our knowledge, this is the first study to investigate the pre-diagnosis stage. By analyzing temporal causality, we further confirmed that retinal changes may precede an ALS diagnosis. Our study also benefits from a relatively large sample size, strict exclusion criteria, and the use of only high-quality images for analysis. Nevertheless, several limitations remain in our study. Firstly, due to the low incidence rate of ALS, it was challenging to achieve complete case confirmation and collect detailed genotype and phenotype data on ALS-related diseases in a specialized manner within large population cohorts. As a result, without detailed clinical information on ALS, we cannot correlate retinal morphology with disease duration or severity. Second, since our study only includes a single pre-diagnosis measurement of OCT data in patients with ALS, rather than dynamic longitudinal data from multiple time points, two issues remain unresolved. On one hand, we cannot further establish whether the thinning of PRL and the thickening of RPE represent early pathological changes during the pre-symptomatic phase of ALS. On the other hand, we are unable to determine the role of retinal indicators in assessing disease progression due to the lack of evidence on retinal changes over the course of ALS. Finally, as this is an observational study, reverse causality cannot be entirely excluded. Although sensitivity analyses excluding short follow-up periods were conducted, due to the long latency, heterogeneity in disease progression, and diagnostic delays in ALS [63], retinal changes observed prior to clinical diagnosis may also indicate alterations in the later stages of the disease. Further research, including interventional studies and mendelian randomization analyses, is needed to clarify causality. Prospective studies with dynamic retinal monitoring throughout ALS progression may help determine the timing, location, and clinical relevance of retinal changes, as well as their potential role in disease assessment and subtype stratification.

In conclusion, a thinner PRL and thicker RPE may precede the clinical diagnosis of ALS, offering potential clues for early diagnosis and insights into the disease's pathogenesis.

## Ethics approval and consent to participate

UK--the North West Multi approved Biobank-Centre Research Ethics Committee (Ref: 11/NW/0382), with all participants providing written informed consent to participate in the UK Biobank study. This research has been conducted using the UK Biobank Resource under Application Number 108832.

## Supporting information

**S1 STROBE Checklist.   Checklist of items that should be included in reports of observational studies.** STROBE, Strengthening the Reporting of Observational Studies in Epidemiology.
(DOCX)

**S1 Text.   Fig A**. The histogram of OCT parameters. **Fig B**. Correlation analysis between retinal markers, ocular diseases, and incident ALS risk. **Table A**. Code information for ocular diseases. **Table B**. Schoenfeld residuals test. **Table C**. Multi-collinearity analysis. **Table D.** Normality test of retinal marker. **Table E.** Associations between retinal markers and incident ALS, additionally adjusted for smoking status. **Table F.** Associations between retinal markers and incident ALS, additionally additionally adjusted for maternal smoking around birth. **Table G.** Associations between retinal markers and incident ALS, excluding individuals with ocular disease at baseline and follow-up. **Table H.** Subgroup analysis. **Table J.** Logistic regression results between PRL and RPE with ocular disease disorders. **Table K.** Cox regression results between baseline ocular disease disorders and incident ALS risk. **Methods.** Obtaining of covariates. **Analysis A.** Relationships between

PRL and RPE. **Analysis B.** OCT parameters and ALS subtype. **Analysis C.** PRL, RPE and time to ALS diagnosis. **Analysis D.** OCT parameters and brain MR images. **Analysis E.** MR analysis of OCT-related parameters and ALS. **Analysis F.** MR analysis of retinal vascular features and ALS. **Analysis G.** Retinal microvascular Changes and ALS.
(PDF)

## Author contributions

**Data curation:** Wenhua Jiang, Haobo Xie, Zhiyang Lin, Yifan Cheng.

**Methodology:** Binbin Deng.

**Project administration:** Binbin Deng.

**Supervision:** Binbin Deng.

**Validation:** Wen Cao, Huan Yu, Dongsheng Fan, Binbin Deng.

**Visualization:** Chunyang Pang.

**Writing – original draft:** Chunyang Pang.

**Writing – review & editing:** Yaojia Li.

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
