## [Editor Report · Decision Letter 0]

Dear Dr Deng,

Thank you for submitting your manuscript entitled "Can we see amyotrophic lateral sclerosis through the eye before it strikes:an optical coherence tomography-based prospective study" for consideration by PLOS Medicine.

Your manuscript has now been evaluated by the PLOS Medicine editorial staff and I am writing to let you know that we would like to send your submission out for external peer review.

Please re-submit your manuscript within two working days, i.e. by Dec 09 2024.

Feel free to email me at atosun@plos.org or us at plosmedicine@plos.org if you have any queries relating to your submission.

Kind regards,

Alexandra Tosun, PhD

Associate Editor

PLOS Medicine

---

## [Decision Letter · Decision Letter 1]

Dear Dr Deng,

Many thanks for submitting your manuscript "Can we see amyotrophic lateral sclerosis through the eye before it strikes:an optical coherence tomography-based prospective study" (PMEDICINE-D-24-04147R1) to PLOS Medicine. The paper has been reviewed by subject experts and a statistician; their comments are included below and can also be accessed here: [LINK]

As you will see, the reviewers highlight the interesting results, but also raise a number of points for clarification. The editors have also raised a number of concerns, which we ask you to address carefully. After discussing the paper with the editorial team, I'm pleased to invite you to revise the paper in response to the reviewers' comments. We plan to send the revised paper to some or all of the original reviewers, and we cannot provide any guarantees at this stage regarding publication.

We ask that you submit your revision by Mar 18 2025. However, if this deadline is not feasible, please contact me by email, and we can discuss a suitable alternative.

Don't hesitate to contact me directly with any questions (atosun@plos.org).

Best regards,

Alexandra

Alexandra Tosun, PhD

Associate Editor

PLOS Medicine

atosun@plos.org

Comments from the editorial team:

* Please remove any language that suggests the study is prospective. The data for the UK Biobank was collected prospectively, but the study you have carried out is retrospective. Please also revise the title accordingly.

* We feel that it may be confusing to a general audience that you refer to retinal thinning throughout the manuscript, suggesting that multiple measurements were taken. We ask you to clarify what 'thinning' refers to, e.g. a widely accepted average measurement or differences within the study cohort. In the absence of temporal assessments, it might be better to refer to a number rather than calling it thinning (an active process). It should be very clear that you are referring to a thinner status compared to others included (if that is the reference) during the single assessment. In this context, please also clarify that thinning cannot be used as a predictor if you do not show a decrease in thickness, or clarify that you are referring to a thinner layer relative to thicker layers in the same cohort at a single time point. Please note that the same issues apply to the word 'thickening', which would be better described as 'thickness'.

* In the discussion, we suggest adding a statement that a prospective trial is needed to show that thinning/thickening of the relevant layers over time is associated with the development of ALS.

* It also seems that you have no data on layer thickness in people who have developed ALS. Please state this more clearly in the limitations section and add the fact that there are no temporal measurements of any ophthalmic parameter.

* In the Results section, please state the number of people who developed ALS (not just a number in person-years).

Comments from the reviewers:

Reviewer #1: Alex McConnachie, Statistical Review

This review looks at the use of statistics in the paper by Pang et al. The paper is an analysis of UK Biobank data, looking at the associations between OCT measurements and incident ALS.

These are generally good. Cox PH models are used (and the PH assumption is checked) to assess associations. Analyses are done without adjustment, and adjusting for a range of potential confounders. People with events at baseline or with less than 2 years of follow-up are excluded, and the sensitivity of results to longer blanking periods are reported. The post-hoc analysis looking at prevalent ocular diseases is interesting.

Each OCT parameter is considered on its own, and three parameters are found to be associated with incident ALS, though only two (PRL and PRE) remain in the sensitivity analysis. Would it make sense to consider a model with both factors included, to see whether they are independent, or measuring the same variation in risk?

The authors choose to "standardise" the OCT parameters before modelling. This is not strictly necessary, but is OK if the authors want to report the hazard ratio per standard deviation increase. All but two parameters are z-transformed, but the results section states that "most variables exhibiting a skewed distribution before standardization". Surely z-transformation won't make any difference to the skew?

Figure 2 reports the distributions of the OCT parameters, which is good to see. However, the first 6 panels show very narrow distributions, so it is hard to tell what the distributions look like. The spline fits are shown for the full range of the x axes. Even if there are a handful of data points far away from the main body of data, these will be quite extreme outliers, so the splines have very little support at the edges of the distributions, and these outliers might be having a large influence on the slopes of the linear trends. Would it make more sense to exclude these extreme outliers in the OCT parameters? Do these people have existing ocular pathology?

The last three panels are much easier to see, but it looks like disc diameter is shown after transformation - readers may be more interested in the distribution of the real data. Table 1 doesn't help - it also reports the transformed values. Seeing the actual data might help in understanding why it was necessary to use an inverse rank normal distribution. The same can be said for VCDR - what is the actual distribution of the data? Also, in the methods, some more detail is necessary - what does "regressed and transformed" actually mean here?

This brings me back to my original comment, that transformation of predictor variables is not necessary in order to fit Cox regression models. Z-transformations are OK (so that the association per SD increase can be reported), but the transformations applied to disc diameter and VCDR (or is it CVDR as in the statistical methods section?) are a little "odd". Would the overall conclusions be any different if simple z-transformations had been applied?

The subgroup analyses are a good feature, but the models appear potentially unstable in some instances, with very wide confidence intervals (both ethnicity analyses, and the hypertension and diabetes analyses for RPE). As a result, I am not sure I trust the ethnicity interaction for PRL. Would it help to give the numbers in each subgroup in Figure 3?

Reviewer #2: Pan and colleagues have conducted an interesting study on the role of OCT as a biomarker for ALS in the prodromal phase. This research is solid, well-designed, and included a large number of participants.

Some suggestions to improve the manuscript:

1. Several years before the cited literature, some researchers discussed the potential biomarker role of retinal alterations in ALS. The manuscript would benefit from including these references (e.g., J. Neurol 2021).

2. Although the authors briefly mention this in the penultimate paragraph of the discussion, it would be interesting to reflect on whether the findings described in the manuscript will have the same biomarker value for the different subtypes of ALS (spinal, bulbar, etc.).

3. Finally, including a section in the discussion attempting to correlate the observed retinal changes with the pathogenesis of ALS would be of great interest.

Reviewer #3: This is a very important paper, meticulously executed, statistically powerful and showing results that suggest early manifestations of ALS in patients' retinas long before they develop other manifestations of neurovegetative disease.

unlike the majority of papers looking at the oculomics of neurogenerative disease which look at retinas of patients with various neurodegenerations after they have become symptomatic or they attempt to correlate disease severity with retinal findings, here the authors have very effectively studied the retinas of patients ultimately affected before they developed clinical manifestations of ALS hence proving that the disease process is affecting the retinas early on and could ultimately proven to be an effective screening tool.

I found both Figures 2 and 4 very difficult to understand and wonder if they can either be eliminated (esp Fig 4) and the information they summarize left to stand alone as summarized in the text.

I assume none of the ALS patients also had new ophthalmic diagnoses that could be affecting PR or RPE layers.

i am not an expert in biostatistics and the paper should be rigorously reviewed to ensure the right methods are being used.

with the assumption that the statistically analyses were properly chosen, can there be a statement beyond OR or a 19 or 20% increased risk respectively such as how many people with thin PRL or thick RPE did not develop ALS and were all of the ALS patients found to have thickness measures that were a least one standard deviation outside of average?

It was a small sample but were the thinnest PRL seen in patients who developed the disease soonest?

did the UK biobank include MRI brain at the same time as initial OCT?, if so would be interesting to see if volume/atrophy correlayed with CT findings and ultimately developing ALS

Consider including these references which are complementary to you introduction and discussion:

1: Fawzi AA, Simonett JM, Purta P, Moss HE, Lowry JL, Deng HX, Siddique N, Sufit

R, Bigio EH, Volpe NJ, Siddique T. Clinicopathologic report of ocular

involvement in ALS patients with C9orf72 mutation. Amyotroph Lateral Scler

Frontotemporal Degener. 2014 Dec;15(7-8):569-80. doi:

10.3109/21678421.2014.951941. Epub 2014 Oct 16. PMID: 25319030; PMCID:

PMC4327840.

2: Volpe NJ, Simonett J, Fawzi AA, Siddique T. Ophthalmic Manifestations of

Amyotrophic Lateral Sclerosis (An American Ophthalmological Society Thesis).

Trans Am Ophthalmol Soc. 2015;113:T12. PMID: 26877563; PMCID: PMC4731009.

Reviewer #4: Summary:

This large-scale prospective study utilizes data from the UK Biobank to explore potential retinal biomarkers predictive of amyotrophic lateral sclerosis (ALS) using Optical Coherence Tomography (OCT). Over a 14.11-year follow-up of 53,824 participants, the study identifies associations between thinner photoreceptor layers (PRL) and thicker retinal pigment epithelium (RPE) with an increased risk of ALS. These findings are significant even after adjusting for confounders and in sensitivity analyses excluding shorter follow-up durations. The study suggests that retinal changes may precede ALS diagnosis, offering insights into early detection and disease pathogenesis.

While the study addresses a novel and potentially impactful question, several methodological concerns, interpretative overstatements, and gaps in the discussion need addressing.

Major Revisions:

1. Causal Inference and Reverse Causality:

o The authors suggest that retinal changes precede ALS diagnosis, implying causality. However, reverse causality cannot be entirely excluded, especially given the long preclinical phase of ALS. Although sensitivity analyses excluding short follow-ups were conducted, a more robust discussion is needed on potential biases and reverse causality.

2. Specificity of Retinal Changes to ALS:

o The findings of PRL thinning and RPE thickening are intriguing but not necessarily specific to ALS. Both alterations have been described in other neurodegenerative conditions, such as Parkinson's disease and Alzheimer's disease.

3. Potential Confounding by Ocular Diseases:

o While the study attempts to adjust for ocular conditions, residual confounding remains a concern, especially since the RPE and PRL are susceptible to age-related changes and other retinal disorders.

4. Smoking as a Modifier:

o The finding that smoking amplifies retinal alterations in ALS is notable. However, smoking is also an independent risk factor for various retinal pathologies, which could confound the association.

5. Biological Plausibility and Mechanistic Insights:

o The discussion on RPE thickening being linked to mitochondrial dysfunction and oxidative stress in ALS is compelling but somewhat speculative.

6. Inclusion of Retinal Vascular Parameters:

o The study focuses on structural OCT metrics but does not consider retinal vascularization, despite emerging evidence of microvascular involvement in ALS associated with disease aggressiviness.

Minor Revisions:

1. Terminology Consistency:

o Ensure consistent use of abbreviations (e.g., PRL, RPE) and terminology throughout the manuscript, especially in figure legends and tables.

2. Statistical Reporting:

o Report exact p-values where possible (e.g., P = 0.002 instead of P < 0.05) for transparency.

o Clarify in the methods whether multiple testing corrections (e.g., Bonferroni) were applied, especially given the number of retinal parameters tested.

3. Clarity in Figures and Tables:

o Figure 2: Improve the readability of the histograms and non-linear trend lines. Consider adding 95% confidence intervals to the plots.

o Table 2: Clearly indicate which variables remained significant after adjustment and in sensitivity analyses using bold formatting or footnotes.

---

* Please upload any figures associated with your paper as individual TIF or EPS files with 300dpi resolution at resubmission; please read our figure guidelines for more information on our requirements: http://journals.plos.org/plosmedicine/s/figures. While revising your submission, please upload your figure files to the PACE digital diagnostic tool, https://pacev2.apexcovantage.com/. PACE helps ensure that figures meet PLOS requirements. To use PACE, you must first register as a user. Then, login and navigate to the UPLOAD tab, where you will find detailed instructions on how to use the tool. If you encounter any issues or have any questions when using PACE, please email us at PLOSMedicine@plos.org.

* FINANCIAL DISCLOSURES: The funding statement should include: specific grant numbers, initials of authors who received each award, URLs to sponsors’ websites. Also, please state whether any sponsors or funders (other than the named authors) played any role in study design, data collection and analysis, the decision to publish, or preparation of the manuscript. If they had no role in the research, include this sentence: “The funders had no role in study design, data collection and analysis, decision to publish, or preparation of the manuscript.”

* COMPETING INTERESTS: All authors must declare their relevant competing interests per the PLOS policy, which can be seen here: https://journals.plos.org/plosmedicine/s/competing-interests

* DATA AVAILABILITY: The Data Availability Statement (DAS) requires revision. PLOS Medicine requires that the de-identified data underlying the specific results in a published article be made available, without restrictions on access, in a public repository or as Supporting Information at the time of article publication, provided it is legal and ethical to do so. Please see the policy at http://journals.plos.org/plosmedicine/s/data-availability and FAQs at http://journals.plos.org/plosmedicine/s/data-availability#loc-faqs-for-data-policy

FIGURES AND TABLES

SUPPLEMENTARY MATERIAL

REFERENCES

STUDY TYPE-SPECIFIC REQUESTS

* Abstract: Please include the study design, population and setting, number of participants, years during which the study took place (enrollment and follow up), length of follow up, and main outcome measures.

* Please ensure that the study is reported according to the STROBE (or appropriate STOBE extension) guideline (available from: https://www.equator-network.org/reporting-guidelines/strobe) and include the completed STROBE (or STROBE extension) checklist as Supporting Information. Please add the following statement, or similar, to the Methods: "This study is reported as per the Strengthening the Reporting of Observational Studies in Epidemiology (STROBE) guideline (S1 Checklist)." When completing the checklist, please use section and paragraph numbers, rather than page numbers.

* For all observational studies, in the manuscript text, please indicate: (1) the specific hypotheses you intended to test, (2) the analytical methods by which you planned to test them, (3) the analyses you actually performed, and (4) when reported analyses differ from those that were planned, transparent explanations for differences that affect the reliability of the study's results. If a reported analysis was performed based on an interesting but unanticipated pattern in the data, please be clear that the analysis was data driven.

* Please state in the Methods section whether the study had a prospective protocol or analysis plan. If a prospective analysis plan (from your funding proposal, IRB or other ethics committee submission, study protocol, or other planning document written before analyzing the data) was used in designing the study, please include the relevant document(s) with your revised manuscript as a Supporting Information file to be published alongside your study and cite it in the Methods section. A legend for this file should be included at the end of your manuscript. If no such document exists, please make sure that the Methods section transparently describes when analyses were planned, and when/why any data-driven changes to analyses took place. Changes in the analysis, including those made in response to peer review comments, should be identified as such in the Methods section of the paper, with rationale.

---

## [Decision Letter · Decision Letter 2]

Dear Dr. Deng,

Thank you very much for re-submitting your manuscript "Can we see amyotrophic lateral sclerosis through the eye before it strikes:an optical coherence tomography-based study" (PMEDICINE-D-24-04147R2) for review by PLOS Medicine.

Thank you for your detailed response to the reviewers' and editors’ comments. I have discussed the paper with my colleagues, and it has also been seen again by three of the original reviewers. The changes made to the paper were satisfactory to the reviewers. As such, we intend to accept the paper for publication, pending your attention to the reviewers' and editors' comments below in a further revision. When submitting your revised paper, please once again include a detailed point-by-point response to the reviewers' and editorial comments.

[LINK]

In revising the manuscript for further consideration here, please ensure you address the specific points made by each reviewer and the editors. In your rebuttal letter you should indicate your response to the reviewers' and editors' comments and the changes you have made in the manuscript. Please submit a clean version of the paper as the main article file. A version with changes marked must also be uploaded as a marked up manuscript file. Please also check the guidelines for revised papers at http://journals.plos.org/plosmedicine/s/revising-your-manuscript for any that apply to your paper.

We ask that you submit your revision within 1 week (May 23 2025). However, if this deadline is not feasible, please contact me by email, and we can discuss a suitable alternative.

Please do not hesitate to contact me directly with any questions (atosun@plos.org). If you reply directly to this message, please be sure to 'Reply All' so your message comes directly to my inbox.

We look forward to receiving the revised manuscript.

Sincerely,

Alexandra Tosun, PhD

Associate Editor

PLOS Medicine

plosmedicine.org

Comments from Reviewers:

Reviewer #1: Alex McConnachie, Statistical Review

I thank Pang and colleagues for their consideration of my original comments, and I am happy with their responses, and the changes to the paper.

One very minor exception is the new text at lines 178 to 182. Personally, I would remove "before standardization" since this will have no impact on skewness.

Reviewer #3: I believe the authors have addressed my and the other reviewers comments in a thoughtful and thorough manner and enhanced the quality of the manuscript

Reviewer #4: I thank the authors for their careful and thorough revision. I found their responses clear, thoughtful, and scientifically sound. The main concerns I raised—particularly regarding causal inference, the interpretation of PRL and RPE, and the absence of vascular OCT parameters—were addressed appropriately. The inclusion of new analyses, clarification of terminology, and validation in a clinical cohort all contribute to strengthening the manuscript. I have no further major concerns and support the revised version for publication.

[LINK]

Requests from Editors:

GENERAL

* Please confirm that your title complies with to PLOS Medicine's style. Your title must be nondeclarative and not a question. It should begin with main concept if possible. "Effect of" should be used only if causality can be inferred, i.e., for an RCT. Please place the study design ("A randomized controlled trial," "A retrospective study," "A modelling study," etc.) in the subtitle (i.e., after a colon).

Suggestion: Analysis of retinal markers and incident amyotrophic lateral sclerosis: an optical coherence tomography-based cohort study

* Statistical reporting: Please revise throughout the manuscript, including tables and figures.

- Please report statistical information as follows to improve clarity for the reader ""22% (95% CI [13,28]; p</=)"".

- Please separate upper and lower bounds with commas instead of hyphens as the latter can be confused with reporting of negative values.

- Please repeat statistical definitions (HR, CI etc.) for each set of parentheses.

* Please ensure that all abbreviations are defined at first use throughout the text (including statistical abbreviations). Please also check figures and tables.

* Please ensure that tables and figures, including those in supplementary files, are appropriately referenced in the main text.

* Financial Disclosure: The funding statement should include: specific grant numbers, initials of authors who received each award, URLs to sponsors’ websites. Also, please state whether any sponsors or funders (other than the named authors) played any role in study design, data collection and analysis, the decision to publish, or preparation of the manuscript. If they had no role in the research, include this sentence: “The funders had no role in study design, data collection and analysis, decision to publish, or preparation of the manuscript.”

* The Data Availability Statement (DAS) requires revision. For each data source used in your study:

* Please revise for use of patient-centered language. Please note that patient-centered language is constructed with the use of post-modified nouns (e.g. 'patients with ALS’ (or similar) instead of ‘ALS patients’) putting the person first in the sentence structure.

* Please ensure that changes in the analysis -- including those made in response to peer review comments -- are identified as such in the Methods section of the paper, with rationale.

ABSTRACT

* Please confirm that your abstract complies with our requirements, including providing all the information relevant to this study type https://journals.plos.org/plosmedicine/s/submission-guidelines#loc-abstract

* Please ensure that all numbers presented in the abstract are present and identical to numbers presented in the main manuscript text.

* l.30, “(year 2006-1010)” – please clarify that 2006-2010 only refers to baseline.

* l.32, we suggest removing ‘prospective’.

* l.33: We think it would be helpful to state how many of the participants developed ALS (as done in the Results section of the main text).

* Please include basic demographics, i.e. mean age, sex, ethnicity/race.

* Please include the important dependent variables that are adjusted for in the analyses.

* Please define the retinal markers investigated.

AUTHOR SUMMARY

* The Author Summary requires revision: Each sub-heading should contain 2-3 single sentence, concise bullet points containing the most salient points from your study. Please revise accordingly. You may want to look at a recent publication to look at the formatting: https://doi.org/10.1371/journal.pmed.1004568

* In the author summary, in the final bullet point of 'What Do These Findings Mean?', please include the main limitations of the study in non-technical language.

METHODS AND RESULTS

* Please check that any use of statistical terms (such as trend or significant) are supported by the data, and if not please remove them. Please note that the term trend should be used only when the test for trend has been conducted.

* l.218: We think it would be useful to briefly mention that you did not observe any associations between RNFL, GCIPL, Macular thickness, Macular volume, Disc diameter and VCDR and ALS risk.

* Figure 1: Please revise the title, e.g. “Participant selection and study design” (or similar).

* Figure 2: Please explain the meaning of the shaded area in the figure description. There’s a typo in “Dis diameter”. Also, the y-axis of the Disc Diameter graph seems lopsided and longer than the others. Please briefly mention the median time between marker assessment and ALS diagnosis.

* Figure 3: We think it would be useful to briefly explain in the figure description what a TDI above or below 2.120 indicates. Please revise the figure title to be more specific. Please include the n numbers for each group.

* Figure 4: Please note that figures should not normally be introduced in the Discussion section. We think it's fine to keep figure 4 in the discussion if you prefer.

* Table 3: Please indicate the number of individuals for each group (i.e., each disease).

General Editorial Requests

---

## [Editor Report · Decision Letter 3]

Dear Dr Deng, 

On behalf of my colleagues and the Guest Academic Editor, Raffaele Dubbioso, I am pleased to inform you that we have agreed to publish your manuscript "Analysis of retinal markers and incident amyotrophic lateral sclerosis: an optical coherence tomography-based cohort study" (PMEDICINE-D-24-04147R3) in PLOS Medicine.

I appreciate your thorough responses to the reviewers' and editors' comments throughout the editorial process. We look forward to publishing your manuscript, and editorially there are only a few remaining points that should be addressed prior to publication. We will carefully check whether the changes have been made. If you have any questions or concerns regarding these final requests, please feel free to contact me at atosun@plos.org.

Please see below the minor points that we request you respond to:

* In the introduction, "ALS patients" is still used twice instead of "patients with ALS." Please revise.

* Figure 3: Please revise the figure title and make it more specific, e.g. “Subgroup analyses for PRL and RPE”.

* Figure 3: Please note that two of the n-numbers (Hypertension, yes; Diabetes, yes) appear to be squished. Please revise.

* The term "trend" is used to refer to a nonsignificant P value. The term trend should be used only when the test for trend has been conducted. Please revise accordingly.

Before your manuscript can be formally accepted you will need to complete some formatting changes, which you will receive in a follow up email (including the editorial requests above). Please be aware that it may take several days for you to receive this email; during this time no action is required by you. Once you have received these formatting requests, please note that your manuscript will not be scheduled for publication until you have made the required changes.

PRESS

Sincerely, 

Alexandra Tosun, PhD 

Associate Editor 

PLOS Medicine